# Development of a Rapid Selection System for Salt-Resistant Mutants of *Nicotiana benthamiana* through Protoplast Culture after Gamma Irradiation

**DOI:** 10.3390/plants9121720

**Published:** 2020-12-07

**Authors:** Da Mon Jin, Seung Hee Choi, Myoung Hui Lee, Eun Yee Jie, Woo Seok Ahn, Su Ji Joo, Joon-Woo Ahn, Yeong Deuk Jo, Sung-Ju Ahn, Suk Weon Kim

**Affiliations:** 1Biological Resource Center, Korea Research Institute of Bioscience and Biotechnology, 181 Ipsingil, Jeongeup-si 580-185, Jeollabuk-do, Korea; damon@kribb.re.kr (D.M.J.); csh@kribb.re.kr (S.H.C.); mhlee17@kribb.re.kr (M.H.L.); jeannie@kribb.re.kr (E.Y.J.); dntjr0412@kribb.re.kr (W.S.A.); suji@kribb.re.kr (S.J.J.); 2Department of Bioenergy Science and Technology, Chonnam National University, Gwangju 500-757, Korea; asjsuse@chonnam.ac.kr (S.-J.A.); 3Department of Applied Plant Science, Chonnam National University, Gwangju 500-757, Korea; 4Radiation Breeding Research Team, Advanced Radiation Technology Institute, Korea Atomic Energy Research Institute, 29, Geumgu-gil, Jeongeup-si 580-185, Jeollabuk-do, Korea; joon@kaeri.re.kr (J.-W.A.); jyd@kaeri.re.kr (Y.D.J.)

**Keywords:** gamma irradiation, protoplast culture, rapid selection, salt-resistant mutant, plant regeneration, *Nicotiana benthamiana*

## Abstract

We aimed to develop a novel technology capable of rapidly selecting mutant plant cell lines. Salt resistance was chosen as a rapid selection trait that is easily applicable to protoplast-derived cell colonies. Mesophyll protoplasts were cultured in a medium supplemented with 0, 50, 100, 150, 200, 250, and 300 mM NaCl. At NaCl concentrations ≥ 100 mM, cell colony formation was strongly inhibited after 4 weeks of culture. Tobacco protoplasts irradiated with 0, 50, 100, 200, and 400 Gy were then cultured to investigate the effects of radiation intensity on cell division. The optimal radiation intensity was 50 Gy. To develop salt-resistant tobacco mutant plants, protoplasts irradiated with 50 Gy were cultured in a medium containing 100 mM NaCl. The efficiency of cell colony formation from these protoplasts was approximately 0.002%. A salt-resistant mutant callus was selected and proliferated in the same medium and then transferred to a shoot inducing medium for adventitious shoot formation. The obtained shoots were then cultured in a medium supplemented with 200 mM NaCl and developed into normal plantlets. This rapid selection technology for generating salt-resistant tobacco mutants will be useful for the development of crop varieties resistant to environmental stresses.

## 1. Introduction

Research into radiation breeding began with various crop plants in the 1960s [1]. Radiation mutant breeding has predominantly focused on crop characteristics that increase productivity, such as those controlling plant growth, early maturation, increased yield, and resistance to disease [2,3]. However, there are several limitations to radiation breeding research. Primarily, random mutations or chimeras are frequently formed after irradiation, and many individuals are needed for traits to be passed from the M1 generation to the next generation for trait fixation, and the selection of mutant strains with fixed traits requires the continuous breeding and selection process to be repeated [4]. In addition, the mutation rate and range of variation produced by irradiation differ with crop species and variety, and this creates insufficient reproducibility. Even if considerable time and labor are invested in selecting mutants with excellent traits, it is difficult to fix these traits owing to continuous separation [4]. Furthermore, the selection of mutants without visible phenotype changes has greater limitations.

Plant protoplasts are cells in which the cell walls have been removed using a cell wall-degrading enzyme [5]. Similar to animal cells, these plant protoplasts can be directly used as a material for gene transfer research and can be regenerated as normal plantlets through cell division and organ formation [6,7]. Calli grown in plant protoplast cultures are genetically homogeneous cell groups formed from a single cell and used as the material for research into the mechanisms related to the differentiation of plant cells [8]. In addition, protoplast cultures allow for numerous cells to be cultured in a small sterile container, and increasing the scale of the cultures is relatively easy. Therefore, if chemical substances (e.g., antibiotics, heavy metals, and metabolite derivatives) are added to the protoplast culture process, it is possible to select various metabolic mutants [9,10]. In addition, it should be possible to select more diverse environmental resistance mutants by changing the protoplast cultures’ environmental conditions (temperature, light, humidity, etc.). Protoplast culture technology is highly likely to be used as a means for plant regeneration and the development of new varieties by combining new mutant selection systems or gene editing technologies [11].

Since the successful isolation of tomato protoplasts by Cocking in 1960 [12], the isolation and culture of protoplasts have been performed in many plants. Plant regeneration from protoplasts has also been reported in horticultural crops [10,13,14,15] and woody plants [16], as well as in Arabidopsis [17]. However, in many other plant species, the protoplast isolation process has not yet been established [18], and if it has, the efficiency of plant regeneration from the protoplasts is low [19]. These difficulties in protoplast isolation and culturing processes are limiting factors that prevent the agricultural use of protoplast culture technology. The protoplast culture system can be used as a selection system for mutant strains that are resistant to salt [10] and disease [9]. However, there have been no studies on the isolation and culture of protoplasts from irradiated plants for the selection and regeneration of salt-resistant mutants. In this study, rapid selection and plant regeneration systems for salt-resistant mutants were established in the protoplast culture stage after irradiation using tobacco as a model plant.

## 2. Results and Discussion

### 2.1. Mitotic Division of Tobacco Protoplast-Derived Cells According to Radiation Intensity

To examine the effects of radiation intensity on mitotic divisions in tobacco protoplast cultures, the frequency of cell aggregates formation was examined at radiation doses of 0, 50, 100, 200, and 400 Gy. Immediately after irradiation, there was little visible difference between the unirradiated and irradiated plants (Figure 1A). However, the frequency of cell aggregates formation according to the radiation intensity was significantly different after protoplast isolation from the irradiated tobacco leaves (Figure 1B). As a result of examining the frequency of cell aggregates formation in the first and second cell division stages after 1 week of protoplast culture, the frequency of cell aggregates formation increased by approximately 5–7.4% in the low-dose radiation treatment group (50, 100 Gy) when compared with that in the control group. When one cell became two cells aggregate and two cells aggregate become four cells aggregate, as observed using a microscope, it was regarded as the first and second cell division stages, respectively. However, the percentage of cell division was greatly reduced in the high-dose radiation treatment groups (200 and 400 Gy), and no cell division occurred in the 400 Gy treatment group (Figure 1B). After 2 weeks of culture, the frequency of cell aggregates formation of the control group was 78.1%, whereas, for the 50, 100, 200, and 400 Gy treatment groups, it had markedly decreased as the radiation intensity increased, resulting in cell divisions of 20.5%, 18.5%, 3.9%, and 0%, respectively (Figure 1B).

The effect of the radiation intensity on the protoplast-derived plant regeneration has not previously been reported in a variety of different plants. Various plant seeds irradiated with high doses of radiation failed to develop their main leaves after germination or died after yellowing [20,21,22,23]. High-dose gamma-ray irradiation induces the production of reactive oxygen species, which causes irreversible damage to plant cells, leading to cell death [24,25]. Furthermore, ionizing radiation can be a potent genotoxic agent as it can directly cause DNA damage, such as strand breaks, abasic sites, oxidized bases, and DNA-protein cross-linking [26]. As direct effects are mediated by direct interaction of the irradiation (IR) with individual DNA, and indirect effects occur via reactive oxygen species (ROS) produced from the molecules surrounding DNA, ionizing radiation can be used as a mutagen in plant breeding [27]. In this study, the frequency of cell aggregates formation was greatly reduced by high-dose radiation treatment (200 and 400 Gy). However, the frequency of cell aggregates formation of the low-dose radiation treatment (50 and 100 Gy) was approximately 1.2 times higher than that of the control group 1 week after the initiation of the culture. By contrast, the frequency of cell aggregates formation decreased by 0.6 times 2 weeks after the initiation of the culture. The exact mechanisms involved in the temporary increase in the frequency of cell aggregates formation by low-dose radiation treatment were not identified. However, the germination rate of *Senna tora* seeds increased by 7.2% when treated with low-dose radiation of less than 100 Gy compared with that of the control group [28]. Similar to the results of previous studies, it was found that the effects of promoting cell division with various weak stresses via low-dose irradiation might have positive effects on the initial cell cultures.

After irradiation of the protoplast cultures, mitotic cell division was observed under an inverted microscope, and after 1 week of culture, cell division occurred in the 50, 100, and 200 Gy radiation-treated cells and the control (Figure 1C). After 5 weeks of culture, cell colonies were formed through cell division in the low-dose treatment groups (50 and 100 Gy) and control, but cell colonies could not be formed at all in the high-dose radiation treatment groups (200 Gy and above), and they turned black and died (Figure 1C). In the 200 Gy radiation treatment group, cell division seemed to be possible with the initial cell cultures, but as the culture period increased, the cells stopped dividing and died. Therefore, for tobacco protoplasts, it would be reasonable to adjust the radiation intensity to less than 100 Gy to form cell colonies through cell division.

### 2.2. Mitotic Division of Tobacco Protoplast-Derived Cells According to NaCl Treatment Concentration

To investigate the effects of the NaCl treatment on the cell division of tobacco protoplast cultures, 0, 50, 100, 150, 200, 250, or 300 mM, NaCl was added to the culture medium. The frequency of cell aggregates formation of protoplasts isolated from tobacco leaves decreased significantly with increasing NaCl treatment concentrations (Figure 2). After NaCl treatment, the divisions in the protoplast-derived cells were observed using an inverted microscope, and after 1 week of culture, cell divisions were observed not only in the control group but also in the 50, 100 mM NaCl treatment group, although the percentage of cell division was lower. (Figure 2A). However, after 5 weeks of culture, cell colonies were formed in the lowest concentration (50 mM) NaCl treatment group and the control but could not be formed at all in the treatment groups with ≥100 mM NaCl, as they turned black and died (Figure 2A).

The early frequency of cell aggregates formation of the protoplast culture was measured (Figure 2B). After 1 week of culture, cell division occurred in all NaCl treatment groups, although the frequency of cell aggregates formation was approximately 5–10% lower than that in the control (Figure 2B). However, after 2 weeks of culture, the frequency of cell aggregates formation in the 0, 50, 100, 150, 200, 250, and 300 mM NaCl treatment groups was 66%, 44.6%, 8%, 0%, 0%, 0%, and 0%, respectively, and no cell division was observed in the treatment groups with ≥150 mM NaCl (Figure 2B). A recent study showed that carrot frequency of cell aggregates formations decreased with >50 mM NaCl treatments and that cells did not divide at all with >200 mM NaCl when cultured to select a salt-resistant cell line from carrot protoplasts [10]. In our study, cell division was possible initially in the 100 mM NaCl treatment group, but as the culture period increased, cell division ceased, and the tobacco cells eventually died. Therefore, 100 mM was the optimal NaCl concentration for the selection of salt-resistant tobacco protoplasts.

### 2.3. Establishment of Optimal Conditions for Selection of a Salt-Resistant Tobacco Radiation Mutant

To select a salt-resistant tobacco mutant, the isolated protoplasts from irradiated tobacco plants were cultured in a NaCl supplement medium to investigate cell division (Figure 3). After irradiation with a radiation intensity of 0, 50, or 100 Gy, cell division was observed in 100 mM NaCl-supplemented medium, and after 1 week of culture, cell division was observed not only in the control but also in the 50 and 100 Gy radiation treatment groups (Figure 3A). However, after 5 weeks of culture, cell colonies were formed through cell division in the 50 Gy radiation treatment group but not in the 100 Gy treatment group (Figure 3A). While cell division was possible in the control cell culture, no cell colonies were formed through cell division in the culture medium containing 100 mM NaCl after 5 weeks of culture (Figure 3A). This result suggests that cell colony formation from tobacco protoplasts is impossible under 100 mM NaCl conditions unless mutations occur. Therefore, cell colonies grown under 100 mM NaCl conditions after irradiation could be selected for mutant strains with salt resistance.

The frequency of cell aggregates formation of tobacco protoplasts was investigated after a combination of irradiation and 100 mM NaCl treatment (Figure 3B). After 1 week of culture, the frequency of cell aggregates formation of the 50 and 100 Gy treatment groups decreased by approximately 18–28% compared with that of the control (Figure 3B). After 2 weeks of culture, the frequency of cell aggregates formation was approximately 100% in both the 50 Gy treatment group and the control, and the 100 Gy treatment group showed a significant decrease in the frequency of cell aggregates formation to 0% (Figure 3B). This result suggests that the formation of cell colonies through cell division is impossible when the 100 Gy irradiation and 100 mM NaCl treatments are combined. We showed that cell colony formation is possible with only the 100 Gy radiation treatment (Figure 1C). Taken together, these results imply that tobacco protoplasts do not form cell colonies through cell division when treated simultaneously with radiation (100 Gy) and NaCl (100 mM) because of excessive stress compared with that in each individual treatment. Therefore, it would be reasonable to adjust the radiation intensity to 50 Gy for the selection of salt-resistant mutants. Subsequently, salt-resistant mutants were selected under the conditions of 50 Gy irradiation and 100 mM NaCl. Thus, the identification of suitable conditions for culturing protoplasts isolated from irradiated plants for the selection of stress-resistant calli is critical. Note that it is especially important to set the appropriate concentration of the selection agent because the protoplast may be damaged by the treatments of the selection agent and irradiation at the same time.

### 2.4. Selection of Salt-Resistant Calli Derived from Tobacco Protoplasts and Their Proliferation

After irradiation (50 Gy), callus proliferation from tobacco protoplast-derived cell colonies and control cell colonies was investigated in a callus induction medium (Figure 4). The protoplast-derived colonies without irradiation all died in the medium supplemented with 100 mM NaCl, and no callus clumps were formed (Figure 4A,B). However, although the protoplasts irradiated with 50 Gy radiation almost died in the medium supplemented with 100 mM NaCl, a few white calli developed (Figure 4C,D). The efficiency of callus formation from tobacco protoplasts obtained by combined irradiation and 100 mM NaCl treatment was approximately 0.002%. The surviving white callus was transferred to a solid medium of the same composition, and shoots were induced for whole plantlet generation. Although the callus formation was at a very low rate, the callus that was formed is expected to be a valuable line as it is more likely to exhibit a salt resistance phenotype. In addition, if the seeds are irradiated and sorted, it requires a lot of time, space, and effort to grow the plants for screening; thus, the advantage of this system is the ability to sort at the callus stage.

### 2.5. Adventitious Shoot Formation of Salt-Resistant Callus Derived from Tobacco Protoplasts

After irradiation, shoot formation was observed from the white callus selected in a medium containing 100 mM NaCl (Figure 5). The green salt-resistant mutant callus (Figure 5A) was derived from tobacco protoplasts grown for approximately 5 weeks in a callus-inducing medium and transferred to a shoot-inducing medium and cultured to induce shoot development. Green spherical leaf primordia developed after approximately 9 weeks of culture (Figure 5B). The induced green protrusion was cut from the callus tissues, transferred to a new medium of the same composition, and cultured for more than 2 weeks. At this time, new green developing leaf material was observed from the protrusion (Figure 5C). After additional culture in a medium of the same composition, the leaves developed into those of normal small plants (Figure 5D). During the development of the small plants, abnormal shoot development occurred at low frequency from the callus. A salt resistance test was conducted using the small plants regenerated into normal plants. The results of this study show that the protoplasts could be regenerated into salt-resistant plants. There have been previous reports that have shown that protoplasts can be isolated from not only tobacco but also carrot [10] plants for their regeneration, and protoplast derived plants have also been generated from *Vitis vinifera* L. [29]. The technology used to regenerate protoplast-derived plants is difficult, but regeneration is being attempted using protoplast culture technology in various crops. However, no studies have reported the regeneration of plants derived from protoplasts using radiation mutation technology. Therefore, it is expected that the method presented in this study could be applied as a novel mutagenesis method if the protoplast culture technology of various crops.

### 2.6. Salt Resistance Test of Young Tobacco Plantlets Derived from Mesophyll Protoplasts

The shoot derived from the salt-resistant mutant was cut and transferred to a medium containing 200 mM NaCl to test salt resistance (Figure 6). Then obtained 0.5 cm shoots from the unirradiated calli and the irradiated calli were transferred to a medium containing 200 mM NaCl. The control plants grew to approximately 0.6 cm after almost 5 weeks of culture in the salt-containing medium, and then stopped (Figure 6A–C). However, the stems derived from the salt-resistant callus proliferated rapidly compared with those in the control group. The stems derived from the salt-resistant callus grew to approximately 2.5 cm after almost 5 weeks of culture, showing a large difference in growth to that of the control (Figure 6D,E). This demonstrated that the selected protoplast-derived mutant strains had salt resistance. Meanwhile, among the selected salt-resistant mutants, there was a mutant that grew about three times faster than the control but had a lower stem growth rate than that of other salt-resistant mutants (Figure 6F). In addition, there were other mutant strains with different growth rates in salt-resistant conditions. It has been reported that the irradiation-induced mutation rate and range of variation differ depending on the crop species and variety [4]. Therefore, there can be differences in salt resistance among the protoplast-derived salt-resistant mutants selected. It is predicted that the mutation variations related to the development of stem differentiation will also vary. In future, to verify the salt resistance of the selected salt-resistant mutant plants, seeds will be harvested and tested for salt resistance.

Here, we combined two technologies, radiation breeding technology, and protoplast cultures, to develop a novel selection system capable of rapidly selecting mutant cell lines in the model plant, tobacco. After irradiation of tobacco plants, a salt-resistant mutant strain could rapidly be selected from the protoplasts, using salt resistance as a selection trait. These results suggest that this system could be a good selection system as this methodology can be applied to develop salt-resistant varieties in other crops. In addition, this methodology could be applied to develop mutant plant strains that are resistant to other environmental and biological stresses by using appropriate selection agents in the protoplast culture stage.

## 3. Materials and Methods 

### 3.1. Plant Materials and In-Vitro Growth Conditions

*Nicotiana benthamiana* seeds were obtained from Dr. Jeong Mee Park of the Korea Research Institute of Bioscience and Biotechnology (KRIBB). The seeds were sterilized for 2 min with 70% ethanol and washed twice with sterile water. The washed tobacco seeds were then surface-sterilized with 1% sodium hypochlorite (commercial *Clorox*) for 10 min and then washed five times using sterile water. The surface-sterilized tobacco seeds were placed on ½ MSBM medium based on the Murashige and Skoog formula [30] composed of ½ MS basal salts, 0.4 mg L^−1^ thiamine-HCI, 100 mg L^−1^ Myo-inositol, 30 g L^−1^ sucrose, and 4 g L^−1^ Gelrite (pH 5.8) for seed germination. Approximately 25 surface-sterilized tobacco seeds were placed in a plastic Petri dish (SPL, 100 × 40 mm) containing approximately 50 mL of the medium. A total of 5 replicates were prepared. Tobacco seeds were incubated for 4 weeks at 25 °C under light culture conditions (light period: 16:8 h, light intensity: 80 µmol m^−2^ s^−1^). The green stem (approximately 1 cm in length) apex was cut from the germinated tobacco plant on a clean bench. The cut stem sections were transferred to a medium (MS0.5K0.1IBA; MS basal salts, 100 mg L^−1^ Myo-inositol, 0.4 mg L^−1^ thiamin, 0.5 mg L^−1^ kinetin, 0.1 mg L^−1^ IBA, 30 g L^−1^ sucrose, 8 g L^−1^ plant agar, pH 5.8) to prepare the plant materials for gamma irradiation. Tobacco stems grown in vitro were transferred to a solid medium of the same composition at approximately 4-week intervals and cultured. Later, immediately after irradiation of the grown tobacco, the isolation and culture of salt-resistant protoplasts were conducted using leaves of tobacco plants grown in vitro.

### 3.2. Isolation and Culture of Protoplasts from Leaf of Nicotiana benthamiana

To isolate protoplasts derived from the irradiated tobacco leaves, leaves (approximately 0.5–1 cm in length) were cut from the tobacco plants that had grown for 4 weeks, and then flame-sterilized scalpels and tweezers were used on a clean bench to cut the leaves at intervals of 1–2 mm. Approximately 40 leaves were placed in a plastic Petri dish (SPL, 90 × 20 mm) containing 10 mL of cell wall-degrading enzyme solution (1% viscozyme, 0.5% celluclast, 0.5% pectin EX (all from Novozyme, 9% mannitol, 3 mM MES, pH 5.8), and then suspension cultures were performed using a stirrer for 6 h (50 rpm, 25 °C, dark culture). After the cell wall-degrading enzyme treatment, the isolated protoplasts were filtered using a stainless-steel mesh (diameter 50 µm and 100 µm) to remove debris. The isolated protoplasts were transferred to a centrifuge tube (14 mL round bottom tube) using a Pasteur pipette and then centrifuged at 800 rpm for 5 min. After removing the supernatant, 10 mL of the protoplast washing solution (W5 [31]; 0.4 g L^−1^ MES, 8.9 g L^−1^ NaCl, 18.3 g L^−1^ CaCl_2_, 0.3 g L^−1^ KCl, pH 5.8) was added to the centrifuge tube containing the protoplasts, re-suspended, and centrifuged for 5 min at 800 rpm. After removing the supernatant, the protoplasts were washed with the same washing solution thrice. Protoplasts recovered after a final washing were transferred into 10 mL of protoplast culture medium (B56I2B0.5N; B5 [32] (Gamborg including vitamins) salts, 60 g L^−1^ myo-inositol, 2 mg L^−1^ BA, 0.5 mg L^-1^ NAA, 20 g L^-1^ sucrose, pH 5.8), re-suspended, and the density was measured using a hemocytometer. The final density of the protoplasts was adjusted to approximately 1 × 10^5^ mL^−1^ using protoplast culture medium and then cultured (2 mL per 60 × 15 mm plastic Petri dish; FALCON) in the dark at 25 °C.

### 3.3. Effect of Radiation Intensity on the Tobacco Plant and Protoplast Cultures

Tobacco plants grown for approximately 4 weeks in vitro were irradiated with high levels of radiation (^60^Co) (Advanced Radiation Technology Institute, high-intensity gamma-ray irradiator). To examine the effects of the radiation intensity on tobacco plants, five treatment groups were irradiated with different radiation intensities (0, 50, 100, 200, and 400 Gy) for 1 h. After irradiation, the death of the plant was checked through visual observation. To investigate the effects of radiation intensity on the cell aggregates formation in protoplast culture, leaves were cut from each treatment area for each radiation intensity immediately after irradiation. Protoplasts were isolated from these leaves, and the frequency of living cell aggregates formation was measured. 

### 3.4. Effect of NaCl Concentrations on Tobacco Protoplast Cultures

The effects of the NaCl concentrations on cell division in tobacco protoplast cultures were examined. For salt treatment, a 5-M NaCl solution was prepared, autoclaved, and then added to each medium. To examine the frequency of cell aggregates formation according to the NaCl treatment concentration, the culture density was adjusted to approximately 1 × 10^5^ mL^−1^ using the protoplasts isolated under the above conditions, and NaCl was added to the protoplast culture medium after 3 days of protoplast culture, and then removed after 8 weeks of protoplast-derived cell culture. A 5-M NaCl solution was added to the protoplast culture medium to produce final concentrations of 0, 50, 100, 150, 200, 250, and 300 mM NaCl. Each treatment was prepared in triplicate. The protoplasts treated with NaCl were cultured under dark culture conditions at 25 °C.

### 3.5. Selection of Salt-Resistant Mutants Derived from Tobacco Protoplasts and Plant Regeneration

Salt-resistant tobacco cell lines were selected from the protoplasts cultured for approximately 4 weeks. The protoplast-derived cell lines were transferred to a plastic Petri dish (SPL, 90 × 20 mm) containing 10 mL of protoplast culture medium (B52B0.5N; B5 (Gamborg including vitamin) salt, 2 mg L^−1^ BA, 0.5 mg L^−1^ NAA, 20 g L^−1^ sucrose, 5.844 g L^−1^ NaCl, pH 5.8) supplemented with 100 mM NaCl and cultured. Protoplast-derived cell line culture conditions were 25 °C, 60 rpm, and light (light period: 16:8 h, light intensity: 80 µmol m^−2^ s^−1^). After 4 weeks, the micro callus was transferred to a solid callus induction medium (B52B0.5N; B5 salt, 2 mg L^−1^ BA, 0.5 mg L^−1^ NAA, 20 g L^−1^ sucrose, 8 g L^−1^ plant agar, pH 5.8) of the same composition but without NaCl and cultured under the same light conditions at 25 °C. The total number of protoplasts was incubated at 1 × 10^5^ mL^−1^, and the efficiency of the survived callus formation was calculated through dividing the number of obtained callus by the total number of protoplasts first cultured. To induce shoot differentiation from the developing green protoplast-derived callus, the callus was transferred to shoot differentiation medium (1/2MS2IAA1BA; 1/2 MS basal salts, 100 mg L^−1^ myo-inositol, 0.4 mg L^−1^ thiamin, 2 mg L^−1^ IAA, 1 mg L^−1^ BA, 30 g L^−1^ sucrose, 8 g L^−1^ plant agar, pH 5.8) and cultured. Ten calli per Petri dish were plated and cultured under light culture conditions (light period: 16:8 h, light intensity: 100–130 µmol m^−2^ s^−1^) at 25 °C. The proliferated green callus was transferred to a solid medium of the same composition every 4 weeks and cultured. Shoots formed from protoplast-derived green callus (approximately 0.5 cm or more in length) were cut and transferred to a plastic Petri dish (SPL, 100 × 40 mm) to which approximately 50 mL of plant growth medium (MS0.5K0.1IBA; MS basal salts, 100 mg L^−1^ myo-inositol, 0.4 mg L^−1^ thiamin, 0.5 mg L^−1^ kinetin, 0.1 mg L^−1^ IBA, 30 g L^−1^ sucrose, 8 g L^−1^ plant agar, pH 5.8) was added and cultured at 25 °C under the same light conditions. 

### 3.6. Salt Resistance Verification for the Tobacco Mutant Plants

The salt-resistance of mutant strain shoots that were derived from the tobacco protoplasts was verified. The formed shoot was cut and transferred to a plant growth medium (MS0.5K0.1IBA; MS basal salts, 100 mg L^−1^ myo-inositol, 0.4 mg L^−1^ thiamin, 0.5 mg L^−1^ kinetin, 0.1 mg L^−1^ IBA, 30 g L^−1^ sucrose, 8 g L^−1^ plant agar, pH 5.8) to which 200 mM NaCl was added, and cultured to confirm the difference in plant growth compared with that of the control. In the unirradiated control group, shoots of the same size were transferred to a medium of the same composition, to which 200 mM NaCl was added, and plant growth was observed. The total number of the observed shoots in the control group and irradiated group were six and nine, respectively, and those were obtained from two replicates. Salt resistance was verified by examining the length of the proliferated stem and the presence or absence of development.

### 3.7. Statistical Analysis

Data analyses were performed using the SPSS software, and the averages with the standard deviations were compared by one-way ANOVA with the Tukey’s test (*p* < 0.05). Different letters in the figures indicate significant differences among the samples at a threshold of *p* < 0.05. Bars in all figures represent means ± SD determined from three biological replicates.

## 4. Conclusions

This study has established a rapid selection system for new mutants through a combination of irradiation and protoplast culture technologies. After irradiation, we were able to rapidly select a salt-resistant tobacco mutant strain from the protoplasts of the model plant tobacco, using salt resistance (NaCl) as a selection agent. This methodology could be applied to a variety of crops that would benefit from the development of salt-resistance. In addition, with the use of different selection agents in the protoplast culture stage, we expect that this method could be used directly to develop mutant strains that are resistant to various environmental and biological stresses such as salt, drought, and cold temperatures.

## Figures and Tables

**Figure 1 plants-09-01720-f001:**
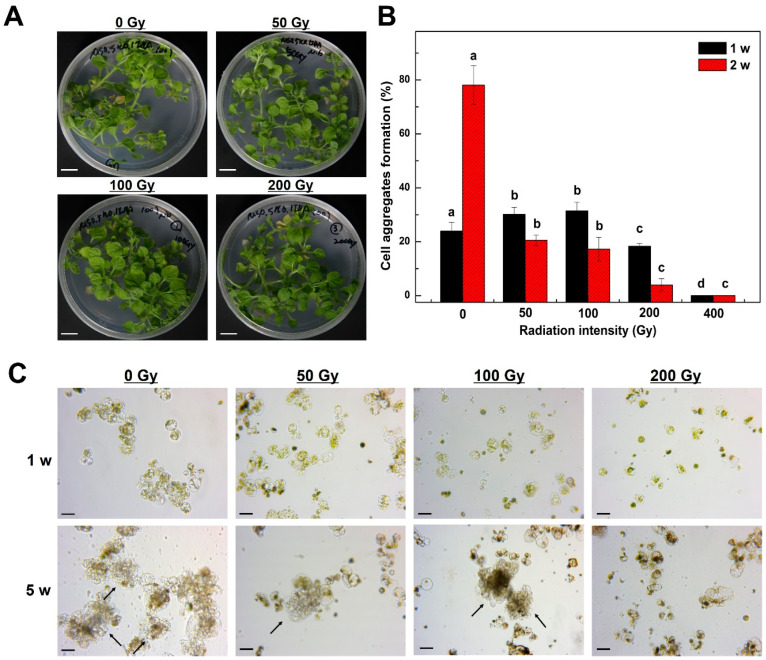
Effect of radiation intensities on tobacco (*Nicotiana benthamiana*) plantlets and frequency of cell aggregates formation from mesophyll protoplast-derived cells. (**A**) Morphological examination of tobacco plantlets after radiation treatment; (**B**) Frequency of cell aggregates formation from mesophyll protoplast-derived cells after radiation treatment. Bars represent means ± SD of three independent experiments. Different letters in (**B**) indicate significant differences among the samples at a threshold of *p* < 0.05 (one-way ANOVA, Tukey’s HSD test); (**C**): Microscopic examination of cell division and cell colony formation from mesophyll protoplast-derived cells after 1 and 5 weeks of incubation. Scale bars represent 1 cm (**A**) and 50 µm (**C**). Arrows indicate cell colonies derived from protoplast cultures. w = week(s); Gy = gray.

**Figure 2 plants-09-01720-f002:**
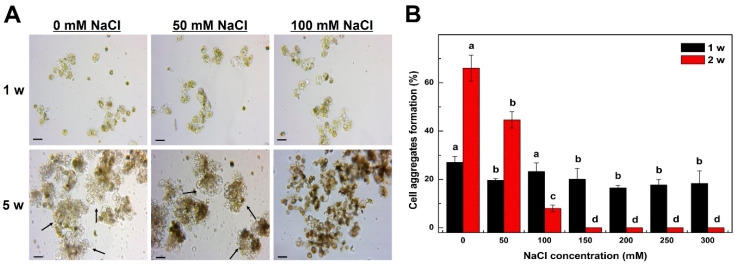
Effect of NaCl concentrations on the frequency of cell aggregates formations from mesophyll protoplast-derived cells of tobacco (*Nicotiana benthamiana*). (**A**): Microscopic examination of cell division and cell colony formation from mesophyll protoplast-derived cells after 1 and 5 weeks of incubation. (**B**): Frequency of cell aggregates formation from mesophyll protoplasts-derived cells after NaCl treatment. Bars represent means ± SD of three independent experiments. Different letters in (**B**) indicate significant differences among the samples at a threshold of *p* < 0.05 (one-way ANOVA, Tukey’s HSD test). Scale bars represent 50 µm (**A**). Arrows indicate cell colonies derived from protoplast cultures. w = week(s).

**Figure 3 plants-09-01720-f003:**
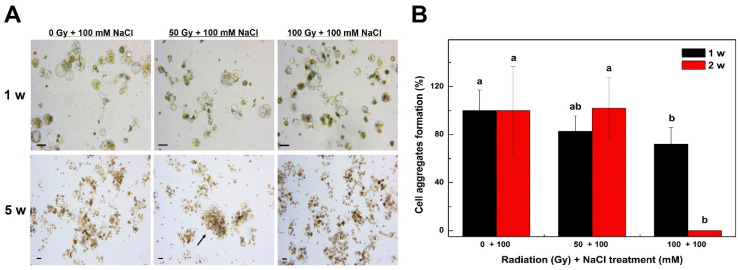
Effects of 100 mM NaCl and different radiation intensities on the frequency of cell aggregates formations from mesophyll protoplast derived cells of tobacco (*Nicotiana benthamiana*). (**A**): Microscopic examination of cell division and cell colony formation from mesophyll protoplast-derived cells after 1 and 5 weeks of incubation. (**B**): Frequency of cell aggregates formation from mesophyll protoplast-derived cells after treatment with NaCl and different radiation intensities. Bars represent means ± SD of three independent experiments. Different letters in (**B**) indicate significant differences among the samples at a threshold of *p* < 0.05 (one-way ANOVA, Tukey’s HSD test). The percentages were calculated against the value of the control. Scale bars represent 50 µm (**A**). Arrows indicate cell colonies derived from protoplast cultures. w = week(s); Gy = gray.

**Figure 4 plants-09-01720-f004:**
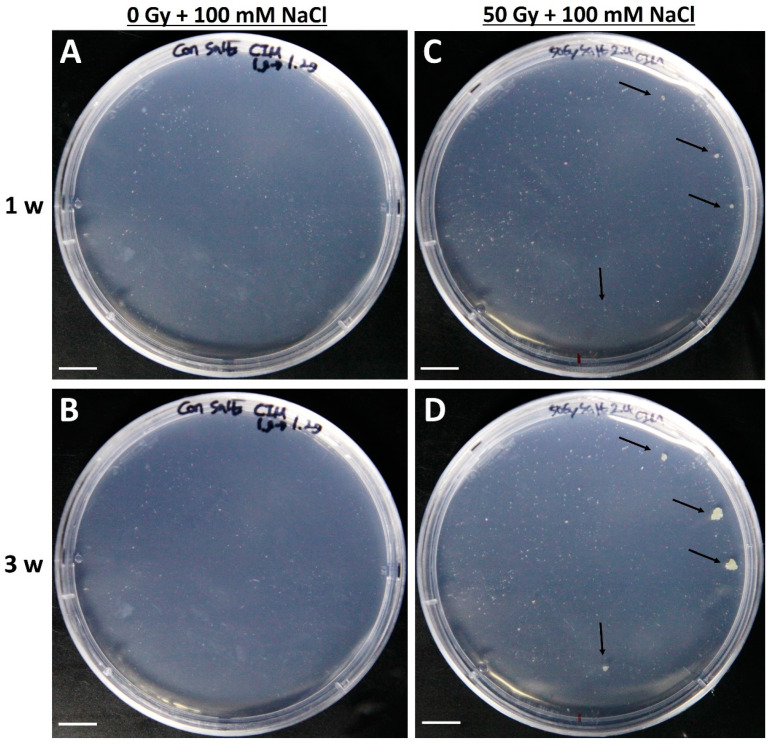
Salt-resistant callus formation from protoplast-derived cell colonies of tobacco (*Nicotiana benthamiana*). (**A**,**B**) No cell colony formation from mesophyll protoplast-derived cells after 1 and 3 weeks of culture on a callus induction medium, respectively. (**C**,**D**) Salt-resistant callus formation from mesophyll protoplast-derived cells of tobacco after 1 and 3 weeks of culture formation on the callus induction medium, respectively. Scale bars represent 1 cm. Arrows indicate salt-resistant calli.

**Figure 5 plants-09-01720-f005:**
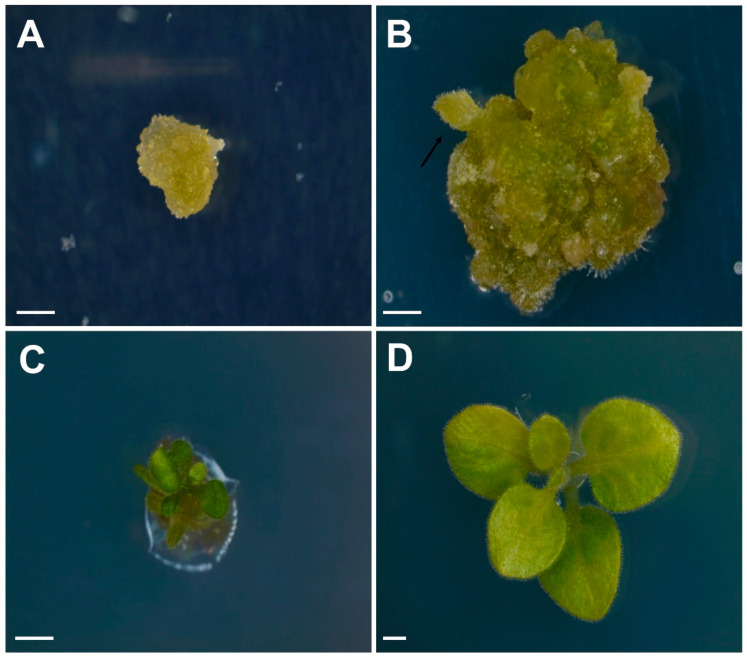
Adventitious shoot formation from protoplast-derived cell colonies of tobacco (*Nicotiana benthamiana*). (**A**) salt-resistant callus from cell colonies; (**B**) Development of green shoot primordia from a green callus; (**C**) Leaf development from shoot primordia; (**D**) Normal plantlet from a salt-resistant callus. Scale bars represent 1 mm in all panels. Arrow indicates green shoot primordia in the callus.

**Figure 6 plants-09-01720-f006:**
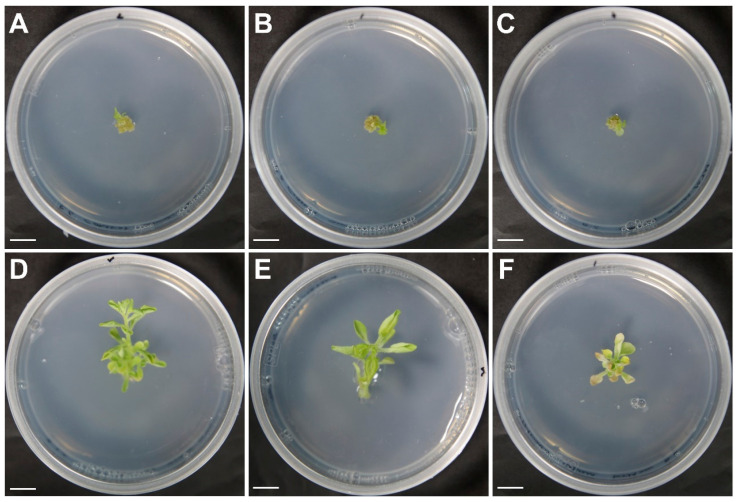
Salt resistance test of young tobacco plantlets derived from mesophyll protoplasts. (**A**–**C**): Control plantlets grown in 200 mM NaCl medium after 5 weeks of incubation; (**D**–**F**) Salt-resistant plantlets grown in 200 mM NaCl medium after 5 weeks incubation. Scale bars represent 1 cm.

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
