# Peer review of "Development of a Rapid Selection System for Salt-Resistant Mutants of Nicotiana benthamiana through Protoplast Culture after Gamma Irradiation"

_plants, 2020, doi:10.3390/plants9121720_

Round 1

Reviewer 1 Report

The article "Development of a Rapid Selection System for Salt-Resistant Mutants of Nicotiana benthamiana through Protoplast Culture after Gamma Irradiation" by Jin et al. (plants-1005026) is a very interesting work concerning the development on mutant lines of tobacco that are salt-tolerant. By means of protoplast culture and the use of ionizing radiation, the authors generate mutants in Nicotiana benthamiana that are able to survive when cultured in a growing media with high concentrations of salt. Despite its interest, there are several issues that prevent me from recommending its publication in its current form.

Major concerns:

- Quality of the images is very low, making it very difficult to appreciate details that are relevant for the evaluation of the work, particularly in the microscopic images. Moreover, this low quality is also present in the Figures 1B, 2B, 3B.

- There is no statistical analysis in Figures 1B, 2B, 3B. This is necessary for the comparisons made.

- There is no such thing as Discussion. Though there are no many references available in this field, a proper Discussion of the results is necessary.

- The authors repeatedly talk about "Division efficiency". This might imply that they analyze every division event and validate if they are successful or not. Do they mean Division rate instead of efficiency? Moreover, in lines 80-81 they talk about "...first and second cell division stages after 1 week...". How do they know these are the first and second divisions? This is not explained.

- The only explanation they give for the generation of the mutant lines is the production of reactive oxygen species (line 91-92). It is expected that ionizing radiation might have other direct effects. In any case, no explanation of how this induces mutations is given.

Other comments:

- English grammar should be improved.

- Text is sometimes repetitive, please check.

- Lines 69-71: sentence should be rewritten.

- Lines 100-102: this sentence makes no sense.

- Lines 125-126: that is not what can be seen in Figure 2A.

- Lines 162-164 and Figure 3B: those percentages are calculated against 100%, but they should be calculated against the value of the control.

- Figure 5: Is the scale bar 1mm in every picture of this Figure?

- The size of the plants is many times given as a direct comparison (1.2 times, 5 times). Proper measures with length units should be given.

- Lines 223-225 and Figure 6: the differences mentioned are not clear in the Figure provided.

- Line 246: "...surface-sterilized tobacco seeds were removed..."??

- Line 254: Did they authors really use 0.8 mg of agar per litre?

- Lines 299-300: Please correct.

- Soil acclimatization: no results are shown concerning this issue.

- Line 339-340: Is salt-resistance the selection marker or the selection agent?

- The origin of the seeds use is not stated.

Overall, draft should be rewritten, quality of the Figures improved, a statistical analysis developed, English should be improved and a proper discussion provided. A more detailed and clearer Materials and Methods section would also be welcome. I encourage the authors to make such changes so their interesting work gets published.

Reviewer 2 Report

Author presents interesting system for mutant selection based on donor plant irradiation and protoplast technology. However, ms has many major and minor problems:

  1. the only well-written section of the article is the introduction;
  2. the others require a thorough content-related and stylistic correction;
  3. the section ‘Materials and methods’ is particularly poorly written, in which a lot of methodological details were omitted (marked in the text), moreover, there is a visible lack of care for the language, numerous repetitions of words and phrases and lack of consistency in the used terminology; section 3.5 was written particularly in a very imprecise and chaotic manner; there is no information about statistical analysis of results (!!!)
  4. the authors use the term ‘cell division efficiency’, but there is no explanation what it means, as well as for term ‘first and second cell division stage’
  5. the results presented in sections 2.3-2.6 were not discussed at all
  6. please note that cell division corresponds to protoplast-derived cells - protoplasts as structures without a cell wall cannot divide, this process takes place after the wall is reconstructed, and then they are not protoplasts but protoplast-derived cells; taking this into account, please correct the entire text;
  7. please improve quality of all figures - current quality is unpublished
  8. other detailed comments were pointed in the text of the ms.

Round 2

Reviewer 1 Report

All my comments were nicely addressed.

Names of the species should always be in italics, including the references.

Author Response

November 30, 2020

The Academic Editor

Plants

Dear Editor:

I appreciate you and the reviewers for taking the time to read and comment on our manuscript. I have enclosed a point-by-point response to the reviewers’ comments for the manuscript (Manuscript No. plants-1005026) entitled as “Development of a Rapid Selection System for Salt-Resistant Mutants of Nicotiana benthamiana through Protoplast Culture after Gamma Irradiation”. The revised text is in blue font to make it easier to identify. I hope that you will find the responses to the comments from the editor and reviewers appropriate and reasonable. I really appreciate you for your time and consideration.

Sincerely,

Suk Weon Kim

Biological Resource Center, Korea Research Institute of Bioscience and Biotechnology (KRIBB)

181 Ipsin-gil, Jeongeup-si,

Jeollabuk-do 56212, Republic of Korea

Phone: 82-63-570-5650

Fax: 82-63-570-5609

REVIEWER COMMENTS (Reviewer 1):

All my comments were nicely addressed.

  1. Names of the species should always be in italics, including the references.

Response 1: As instructed, this has now been revised.

Reviewer 2 Report

The Authors made corrections, but the text still needs to be improved as suggested below:

  1. Author observed not ‘cell division frequency’ rather ‘frequency of cell aggregates formation’ – please change accordingly throughout the text and further clarify in M&M subsection 3.3 (line 330)
  2. line 83: phrase ‘When one cell became two cells and two cells become four cells, ..’ change into ‘When one cell became two cells aggregate and two cells aggregate become four cells aggregate, ..’
  3. in the legend to the figures 1-3 add explanation what show whiskers on the bars
  4. line 306: phrase ‘(1% viscozyme, 0.5% celluclast, 0.5% pectin EX; Novozyme,..’ into ‘(1% viscozyme, 0.5% celluclast, 0.5% pectin EX [all from Novozyme],..’
  5. line 337 - the phrase: ‘then treated 3 days after protoplast isolation’ is not clear, please precise when exactly selecting agent (NaCl) was added to the culture - NaCl was added for 3 days to culture medium just after protoplast isolation and then removed? or the scheme of NaCl treatment was other?
  6. line 366: instead ‘mutant strain shoot that was’ use ‘mutant strain shoots that were’ and then
  7. lines 378-381: please clarify if the variability of the data was expressed by standard deviation (l. 378-379) or standard error SE (l. 381) – now you mix these two statistics.
